# Rethinking emergency risk assessment: A single-center look at shock index and its variants in COVID-19

Annyi Tatiana Belalcazar[1☉], Valeria Monroy Lasso[1,2☉], José Darío Álvarez Herazo[1,2☉], Ana Clarete [iD][2,3‡]*, Roger Figueroa-Paz [iD][1,3‡], Duban Maya-Portillo[2,5☉], Julio Diez-Sepúlveda[1,2,4,5‡]

**1** Facultad de Ciencias de la Salud, Universidad Icesi, Cali, Colombia, **2** Semillero de Investigación en Medicina de Emergencias y Reanimación (SIMER), Facultad de Ciencias de la Salud, Universidad Icesi, Cali, Colombia, **3** Centro de Investigaciones Clínicas, Fundación Valle del Lili, Cali, Colombia, **4** Servicio de Urgencias, Fundación Valle del Lili, Cali, Colombia, **5** Departamento de Cuidados Intensivos, Fundación Valle del Lili, Cali, Colombia

☉ These authors contributed equally to this work.
‡ These authors contributed equally to this work.
* ana.clarete@fvl.org.co

## Abstract

### Background

The Shock Index (SI) is a validated prognostic tool in conditions such as severe trauma and obstetric hemorrhage. During the COVID-19 pandemic, it was used to identify patients at higher risk of clinical deterioration, but results have been inconsistent. This study aimed to evaluate the prognostic value of the SI and its variants in predicting mortality, need for mechanical ventilation, and hospital length of stay in patients with moderate COVID-19.

### Methods and findings

This longitudinal analytical observational study was conducted at a high-complexity hospital in southwestern Colombia and included adults over 18 years of age with moderate COVID-19 treated between 2020 and 2022, using data from the institutional RECOVID registry. A total of 283 patients were analyzed (median age: 61 years; 58.7% male), with cardiovascular and renal comorbidities being predominant. On admission, vital signs were stable (NEWS2: 3.0; shock index: 0.7). ICU admission was required in 29.3% of cases, and overall mortality was 12%. ROC curves and diagnostic accuracy parameters were used to assess the discriminatory ability of the SI and its variants. Most SI variants showed low discriminatory power (AUC < 0.58), except for the estimated shock index and the heart rate–adjusted shock index for mortality, with AUCs of 0.70 and 0.68, respectively, and positive predictive values exceeding 93%.

**Data availability statement:** Due to the data containing personal, sensitive data, the data used in this study can be made available upon reasonable request to researchers who meet the criteria for access to confidential data. Requests should be submitted in writing to the Research Ethics Committee of Fundación Valle del Lili: Comité de Ética en Investigación Biomédica, Fundación Valle del Lili Email: cei@fvl.org.co.

**Funding:** The author(s) received no specific funding for this work.

**Competing interests:** The authors have declared that no competing interests exist.

## Conclusions

Early identification of patients at risk for complications in moderate COVID-19 is essential for optimizing hospital resources. The shock index and its variants showed limited utility as standalone predictors for mortality, ICU admission, and hospital length of stay. Combining SI with other clinical parameters may offer some benefit, but heterogeneity limits generalizability. Future studies should develop and prospectively validate multivariable models integrating clinical, laboratory, and imaging biomarkers.

## Introduction

The COVID-19 pandemic, declared in March 2020, has remained a global health threat. Between 2020 and 2021, COVID-19 became the third leading cause of death in the United States, and to date, it has been associated with more than 6.9 million deaths worldwide [1,2]. Although the World Health Organization declared in May 2023 that COVID-19 no longer constituted a Public Health Emergency of International Concern, the disease continues to pose a threat that requires ongoing surveillance. In Colombia, according to the latest 2025 report from the National Institute of Health, there have been 6,415,764 confirmed cases since the onset of the pandemic, with a case fatality rate of 2.24%. Notably, the burden has been disproportionately high among individuals over the age of 60 [3], highlighting its persistent impact on public health.

A wide range of predictive models have been used during the pandemic to estimate the risk of deterioration, including severity scores originally developed for critical illness, such as APACHE II and SOFA, which, although not designed specifically for COVID-19, demonstrated reasonable prognostic value in this population. A recent systematic review by Appel et al. reported that 242 scores for COVID-19 outcome prognosis have been developed, though most exhibited considerable risk of bias or insufficient validation. Among these, the 4C Mortality Score emerged as the most consistently validated tool, integrating clinical, physiological, and laboratory variables and demonstrating robust predictive performance across multiple populations [4].

In addition, Marcolino et al. developed and externally validated the $ABC_2$-SPH risk score, which also incorporates clinical parameters, including heart rate and blood pressure, alongside physiological and laboratory markers, and demonstrated strong discriminative ability for in-hospital mortality in cohorts from Brazil and Spain [5].

Despite this, the shock index (SI) has been rarely incorporated into these models, despite its long-standing role as a rapid and reliable bedside marker of hemodynamic compromise in trauma, sepsis, and obstetric emergencies. Its simplicity and immediate availability make it particularly valuable in scenarios requiring swift clinical assessment, including patients presenting with COVID-19. Although most studies suggest favorable performance of the SI in patients with COVID-19 [6–8], the evidence remains inconsistent (9), with reports ranging from poor predictive capacity to excellent discrimination of outcomes such as intensive care unit (ICU) admission and mortality [9,10]

Several variants of the SI have been previously developed to improve its performance across different clinical settings by incorporating additional physiological parameters. The modified SI (mSI) replaces systolic blood pressure with mean arterial pressure, the diastolic SI (dSI) incorporates diastolic pressure, the estimated SI (eSI) adjusts for vascular resistance via pulse pressure, and the hypoxia-adjusted SI (HaSI) integrates oxygen saturation to account for respiratory compromise. Evidence from clinical studies indicates that mSI and dSI have demonstrated potential improvement in predictive performance over SI in certain populations, such as patients with trauma, sepsis, or cardiogenic shock [11–15]. Current evidence regarding the prognostic value of SI and its variants in COVID-19 remains controversial, as studies have reported inconsistent predictive performance. Moreover, evidence from Latin American cohorts is limited, making it difficult to generalize existing findings to settings with distinct demographic and comorbidity profiles. To address these gaps, we comprehensively evaluated the prognostic performance of SI and its variants in predicting hospital mortality and ICU admission, as well as the need for invasive mechanical ventilation, among patients with moderate COVID-19 treated at a tertiary care center in a middle-income country.

## Materials and methods

### Study design

A retrospective longitudinal observational study was conducted to evaluate the prognostic performance of the shock index (SI) and its variants in predicting adverse outcomes among patients with moderate COVID-19.

### Setting and population

The study was carried out at a high-complexity hospital in Colombia. We included symptomatic adult patients (aged ≥18 years) with a confirmed diagnosis of moderate COVID-19 by molecular testing, treated between March 2020 and April 2022. The definition of moderate illness was based on the National Institutes of Health (NIH) criteria, which include individuals who show evidence of lower respiratory disease on clinical assessment or imaging and who have an oxygen saturation measured by pulse oximetry ≥94% on room air at sea level [16]. Exclusion criteria included pregnancy and incomplete clinical information (incomplete assessment or absence of imaging studies).

### Patient identification and selection

Eligible patients were identified through the institutional COVID-19 registry (RECOVID), which is implemented in REDCap. The registry includes retrospective data extracted from electronic medical records and institutional applications containing clinical, laboratory, and imaging information. Patients were filtered using ICD-10 codes for COVID-19 and NIH severity criteria. Two trained reviewers independently verified eligibility and extracted data using a standardized protocol. Discrepancies were resolved by consensus

### Variables and outcomes

The following variables were collected:

- Demographic: age, sex.

- Clinical: comorbidities, vital signs at admission (heart rate, systolic and diastolic blood pressure, mean arterial pressure, oxygen saturation).

- Outcomes:

  - ICU admission: proportion of patients admitted to the ICU after initial assessment.

  - Hospital length of stay: number of calendar days from hospital admission to discharge.

  - In-hospital mortality: death occurring during hospitalization

### Data quality and bias control

To minimize information bias due to incomplete entries or variability in data quality, all data underwent validation by personnel trained in data quality assurance. When inconsistencies or missing data were detected, a supplementary review of the corresponding electronic medical records was performed. Cases with missing key variables (HR, SBP, $SpO_2$) were excluded; no imputation was performed.

### Study size

No formal sample size calculation was performed. All patient records in the RECOVID registry meeting the eligibility criteria during the study period were included.

### Statistical analysis

A descriptive analysis of clinical and demographic variables was performed. The normality of continuous variables was assessed using the Shapiro–Wilk test ($p \leq 0.05$). Normally distributed data were reported as mean and standard deviation, while non-normally distributed data were expressed as median and interquartile range (IQR). Categorical variables were presented as absolute frequencies and proportions.

To assess the discriminative ability of SI and its variants in predicting clinical outcomes, receiver operating characteristic (ROC) curves were constructed, and the area under the curve (AUC) was calculated. The optimal cutoff point was determined using Youden's index. Diagnostic performance metrics included positive and negative agreement, positive predictive value (PPV), negative predictive value (NPV), and positive and negative likelihood ratios (LR+ and LR−). All statistical analyses were performed using STATA software (StataCorp, College Station, TX, USA).

### Ethical considerations

The Biomedical Research Ethics Committee (IRB) approved this study (Act No. 10/2024, approved on May 15, 2024; approval letter No. 309). This study complies with the Declaration of Helsinki principles for medical research involving human subjects. According to Resolution 8430 of 1993 issued by the Colombian Ministry of Health, the study was classified as no risk; therefore, informed consent was waived.

## Results

A total of 283 patients were included (Fig 1), with a median age of 61 years (IQR: 48.0–75.0). The majority were male (58.7%). Regarding cardiovascular comorbidities, 51.9% had cardiac disease, 49.1% had hypertension, and 7.1% had coronary artery disease. Only 5.7% had a history of heart failure. Other conditions, such as atrial fibrillation (5.3%), atrioventricular block (0.4%), or ventricular arrhythmias (0.4%), were uncommon. Concerning pulmonary diseases, 9.5% had chronic lung disease, including COPD (4.9%), asthma (1.4%), and pulmonary fibrosis (0.7%). No cases of interstitial lung disease, cirrhosis, or hepatitis were reported. Chronic kidney disease was present in 21% of patients, primarily in stages 3–5, with stage 5 being the most frequent (11.0%). Diabetes mellitus was identified in 21.2% (7.4% non-insulin-dependent and 13.8% insulin-dependent), although data were unavailable for 78.8% of the cohort. A total of 14.5% had a cancer diagnosis, with hematologic malignancies (29.3%) and prostate cancer (19.5%) being the most common types (Table 1).

At admission, vital signs showed a mean systolic blood pressure of 123.0 mmHg, a heart rate of 89 bpm, and oxygen saturation of 96%. All patients had a Glasgow Coma Scale score of 15, and the NEWS2 score had a median of 3.0 (IQR: 1.0–5.0). The shock index had a median value of 0.7 (IQR: 0.6–0.9). A qSOFA score ≥2 was observed in 6.8% of patients. Inflammatory markers included leukocytes at 6,690 (IQR: 5,070–9,925), neutrophils at 4,800 (3,365–7,965), lymphocytes at 1,000 (695–1,470), and a neutrophil-to-lymphocyte ratio (NLR) of 5.1 (2.7–9.5) (Table 2).

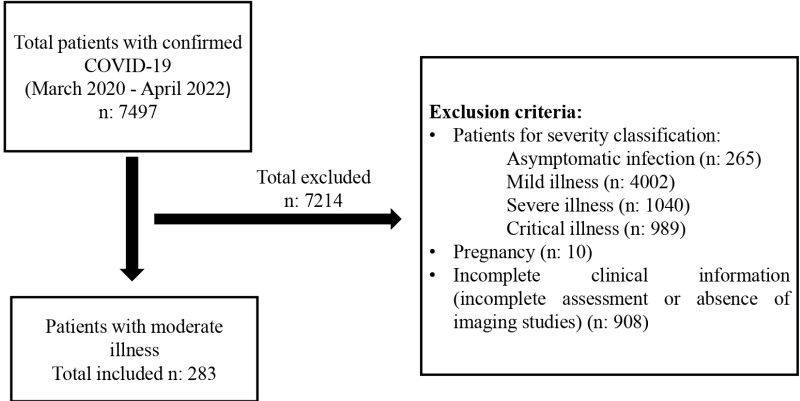

**Fig 1. Patient selection flowchart.** Inclusion and exclusion criteria of the patients.

Regarding complications, 23.3% developed acute respiratory distress syndrome, 9% had a high-risk NEWS2 score, and 29.3% required ICU admission, with a median stay of 8 days (IQR: 3–18.5). Invasive mechanical ventilation was administered to 12% of patients, with a median duration of 10.5 days (IQR: 7.0–17.75) (Table 2).

The median length of hospital stay was 7 days (IQR: 4–14). At discharge, 88% of patients were discharged alive, while 12% died. The most common discharge destination was other (73.1%), followed by homecare (11.7%) (Table 2).

### Diagnostic performance analysis of shock variants

The performance of four shock index variants was compared to predict three clinical outcomes: in-hospital mortality, ICU admission, and hospital stay. The evaluated indices were:

• mSI

• dSI

• eSI

• HaSI

### ICU admission

None of the indices demonstrated acceptable discriminative ability, with AUC values ranging from 0.495 to 0.514 (Table 3). The highest sensitivity (56.6%) was observed with mSI, while eSI showed the highest specificity (60.0%) and negative predictive value (NPV, 73.2%). Positive likelihood ratios (LR+) were low across all indices (range: 1.13–1.21), and the Youden index was below 0.10 in all cases (Table 3) (Fig 2).

### Hospital length of stay

The eSI showed the best overall performance (AUC = 0.574) (Fig 2), with a specificity of 69.2%, NPV of 79.1%, and LR+ of 1.60. The highest sensitivity was observed with HaSI (84.0%), at the cost of low specificity (33.7%). The highest Youden indices were recorded for dSI (0.197) and eSI (0.185) (Table 3) (Fig 3).

### Discharge status (mortality)

The eSI demonstrated the best performance in predicting mortality, with an AUC of 0.699 (Fig 3), sensitivity of 59.0%, specificity of 73.5%, positive predictive value (PPV) of 94.2%, and NPV of 19.7%, with an LR+ of 2.23. HaSI followed

**Table 1. Sociodemographic and clinical characteristics.**

| Variable | n = 283 |
|---|---|
| **Age** | 61.0 (48.0, 75.0) |
| **Gender** | |
| Female | 117 (41.3%) |
| Male | 166 (58.7%) |
| **Cardiovascular disease** | 147 (51.9%) |
| **Arterial hypertension** | 139 (49.1%) |
| **Coronary artery disease** | 20 (7.1%) |
| **Heart failure** | 16 (5.7%) |
| **Pulmonary hypertension** | 3 (1.1%) |
| **Arrhythmias** | 20 (7.1%) |
| **Atrial fibrillation** | 15 (5.3%) |
| **AV block** | 1 (0.4%) |
| **Ventricular arrhythmia** | 1 (0.4%) |
| **Other cardiovascular diseases** | 3 (1.1%) |
| Cardiomyopathy | 1 (0.4%) |
| Valvulopathy | 2 (0.7%) |
| **Pulmonary disease** | 27 (9.5%) |
| **COPD** | 14 (4.9%) |
| **Asthma** | 4 (1.4%) |
| **Pulmonary fibrosis** | 2 (0.7%) |
| **Interstitial lung disease** | 0 (0.0%) |
| **Cirrhosis** | 0 (0.0%) |
| **Hepatitis** | 0 (0.0%) |
| **Chronic kidney disease** | 59 (20.8%) |
| Stage 1 | 1 (0.4%) |
| Stage 2 | 3 (1.1%) |
| Stage 3 | 4 (1.4%) |
| Stage 4 | 2 (0.7%) |
| Stage 5 | 31 (11.0%) |
| Unclassified | 18 (6.4%) |
| **Venous thromboembolic disease** | 2 (0.7%) |
| **Diabetes mellitus** | 60 (21.2%) |
| Non-insulin-dependent | 21 (7.4%) |
| Insulin-dependen | 39 (13.8%) |
| **Cancer** | 41 (14.5%) |
| Breast | 4 (9.8%) |
| Bladder | 3 (7.3%) |
| Renal | 1 (2.4%) |
| Hepatocellular carcinoma | 1 (2.4%) |
| Other | 5 (12.2%) |
| Head and neck | 1 (2.4%) |
| Soft tissue sarcoma | 1 (2.4%) |
| Prostate | 8 (19.5%) |
| Lung | 4 (9.8%) |
| Colorectal | 1 (2.4%) |
| Hematologic | 12 (29.3%) |

*(Continued)*

**Table 1.** (Continued)

| Variable | n = 283 |
|---|---|
| **Cancer treatment** | 37 (13.1%) |
| No current treatment (Remission) | 11 (29.7%) |
| Chemotherapy | 13 (35.1%) |
| Other | 8 (21.6%) |
| Not reported | 5 (13.5%) |

closely, with an AUC of 0.681 and comparable diagnostic performance. The mSI had the highest specificity (91.2%) but very low sensitivity (21.3%) (Table 3) (Fig 4).

## Discussion

This study evaluated the utility of the shock index and its variants in predicting clinical outcomes in patients with moderate COVID-19, including the need for inpatient management (ICU admission, hospital stay), and in-hospital mortality. The findings show that the traditional shock index demonstrated limited discriminative capacity, with areas under the curve (AUCs) of 0.51 for ICU admission, 0.56 for hospital length of stay, and 0.52 for mortality.

When exploring variations of the index, including the use of mean arterial pressure, age, and oxygen saturation, no substantial improvement in performance was observed, except for the HaSI index, which achieved an AUC of 0.70 for mortality prediction, indicating acceptable performance in that context.

Although some predictive models, such as NEWS2 and qSOFA, show favorable performance in specific parameters, their overall usefulness in early risk stratification of patients with COVID-19 remains limited. NEWS2 demonstrates relatively higher sensitivity compared with other scores; however, its values generally remain below 70%, meaning that a substantial proportion of at-risk patients may still be missed. In contrast, qSOFA exhibits very high specificity, approaching 97% for outcomes such as in-hospital mortality, ICU admission, need for mechanical ventilation, and vasopressor use, indicating a strong ability to correctly identify patients unlikely to deteriorate. Nevertheless, the limited sensitivity and modest ability to rule out adverse outcomes observed across these models reduce their reliability for confidently excluding risk. Taken together, these trade-offs underscore the challenges of depending on these tools to identify patients who are truly low risk at emergency department presentation [17].

Our findings are consistent with studies such as that by Van Rensen et al., who reported very low sensitivity (12.3%) for SI in predicting mortality among hospitalized COVID-19 patients, despite high specificity (93.6%) and limited PPV (42.4%). In their analysis, using a low SI cutoff (>0.57)—within the normal range—did not improve its performance in predicting ICU admission. In this context, SI was not useful for early identification of clinical deterioration in hospitalized patients [18]. Other studies have explored modified versions of the shock index. In the study by Trichur et al., the use of SI adjusted for mean arterial pressure was significantly associated with mortality (p = 0.04), whereas the traditional SI was not (p = 0.205), suggesting that certain modifications may improve predictive performance [19].

These variable performances across studies can be attributed to differences in disease severity, patient populations, and the choice of cutoff values. Van Rensen et al.'s study included a broader range of hospitalized patients, while our cohort focused specifically on moderate COVID-19 cases, where more subtle hemodynamic alterations may not be captured by traditional SI. Additionally, their use of a threshold within the normal range inherently limits sensitivity. The superiority of modified indices, such as the SI adjusted for mean arterial pressure in Trichur's study and the eSI in our analysis (AUC 0.699 for mortality), suggests that incorporating additional hemodynamic parameters better reflects the complex pathophysiology of COVID-19-related deterioration. However, even the best-performing index in our study showed only modest discriminative ability for mortality and poor performance for ICU admission and hospital length of stay

**Table 2. Admission physiological variables and clinical outcomes.**

| Variable | n = 283 |
|---|---|
| **Systolic blood pressure** | 123.0 (109.0, 140.0) |
| **Diastolic blood pressure** | 73.7 (15.1) |
| **Pulse pressure** | 49.0 (38.0, 61.0) |
| **Mean arterial pressure** | 90.9 (16.3) |
| **Heart rate** | 89.0 (80.0, 104.0) |
| **Respiratory rate** | 20.0 (18.0, 24.0) |
| **Oxygen saturation** | 96.0 (95.0, 98.0) |
| **Temperature** | 36.5 (36.0, 37.2) |
| **Glasgow coma scale** | 15.0 (15.0, 15.0) |
| **Respiratory distress** | 48 (17.0%) |
| **Shock index** | 0.7 (0.6, 0.9) |
| **NEWS2 score** | 3.0 (1.0, 5.0) |
| **NEWS classification** | |
| Low | 193 (68.2%) |
| Medium | 64 (22.6%) |
| High | 26 (9.2%) |
| **Leukocytes** | 6,690.0 (5,070.0, 9,925.0) |
| **Neutrophils** | 4,800.0 (3,365.0, 7,965.0) |
| **Lymphocytes** | 1,000.0 (695.0, 1,470.0) |
| **INL index** | 5.1 (2.7, 9.5) |
| **Eosinophils** | 0.0 (0.0, 30.0) |
| **Monocytes** | 500.0 (340.0, 725.0) |
| **ILM index** | 2.2 (1.3, 3.3) |
| **Hemoglobin** | 13.4 (11.8, 14.8) |
| **Platelets** | 227,000.0 (172,000.0, 294,500.0) |
| **Acute respiratory distress syndrome** | 66 (23.3%) |
| **Sepsis** | 25 (8.9%) |
| **qSOFA on admission** | |
| 0 | 140 (50.0%) |
| 1 | 121 (43.2%) |
| 2 | 19 (6.8%) |
| **APACHE II on admission** | 8.0 (6.0, 14.0) |
| **Ventilator-associated pneumonia** | 3 (1.1%) |
| **Renal replacement therapy** | 13 (4.6%) |
| **Myocarditis** | 0 (0.0%) |
| **Invasive mechanical ventilation** | 34 (12.0%) |
| **Days of invasive mechanical ventilation** | 10.5 (7.0, 17.75) |
| **Days of renal replacement therapy** | 0.0 (0.0%) |
| **Days of vasopressor support** | 3.0 (1.75, 7.0) |
| **Days of inotropic support** | 2.0 (1.0, 5.0) |
| **ICU requirement** | 83 (29.3%) |
| **ICU length of stay** | 8.0 (3.0, 18.5) |
| **Hospital length of stay** | 208 (73.5%) |
| **Discharge status** | |
| Deceased | 34 (12.0%) |
| Alive | 249 (88.0%) |

*(Continued)*

**Table 2.** (Continued)

| Variable | n = 283 |
|---|---|
| **Final disposition** | |
| Homecare | 33 (11.7%) |
| Referral | 9 (3.2%) |
| Other | 207 (73.1%) |

**Table 3. Diagnostic performance of shock index variants.**

| Outcome | Index | AUC | Sensitivity (%) | Specificity (%) | PPV (%) | NPV (%) | LR+ | LR- |
|---|---|---|---|---|---|---|---|---|
| **ICU admission** | mSI | 0.506 | 56.6 | 53.5 | 33.6 | **74.8** | 1.22 | 0.81 |
| | dSI | 0.496 | 44.6 | 60.5 | 31.9 | **72.5** | 1.13 | 0.91 |
| | eSI | 0.505 | 47.0 | 60.0 | 32.8 | **73.2** | 1.18 | 0.88 |
| | HaSI | 0.514 | 54.2 | 53.5 | 32.6 | **73.8** | 1.17 | 0.86 |
| **Hospital length of stay** | mSI | 0.561 | 52.0 | 61.5 | 32.8 | **78.0** | 1.35 | 0.78 |
| | dSI | 0.567 | 58.7 | 61.1 | 35.2 | **80.4** | 1.51 | 0.68 |
| | eSI | 0.575 | 49.3 | 69.2 | 36.6 | **79.1** | 1.60 | 0.73 |
| | HaSI | 0.574 | **84.0** | 33.7 | 31.3 | **85.4** | 1.27 | 0.48 |
| **In-hospital mortality** | mSI | 0.527 | 21.3 | 91.2 | **94.6** | 13.7 | **2.41** | 0.86 |
| | dSI | 0.507 | 57.0 | 52.9 | **89.9** | 14.4 | 1.21 | 0.81 |
| | eSI | **0.700** | 59.0 | 73.5 | **94.2** | 19.7 | **2.23** | 0.56 |
| | HaSI | **0.682** | 54.6 | 73.5 | **93.8** | 18.1 | **2.06** | 0.61 |

(AUC 0.574), reinforcing the need for multiparametric approaches rather than reliance on single hemodynamic indices for risk stratification in moderate COVID-19.

Similarly, studies in older adults have shown results in line with ours. One analysis of elderly patients with COVID-19 reported low AUCs for SI (0.590) and modified SI (0.608) in predicting mortality [20]. Furthermore, there was no strong correlation with other outcomes such as ICU admission, mechanical ventilation, or the development of shock, which aligns with our results showing poor performance for ICU admission across all shock index variants. Consistently, Doğanay et al. reported that mortality increased significantly with age (26.9% in patients <56 years vs. 91.4% in those >77 years), and that oxygen saturation was a critical factor, as no patient with $SpO_2 > 95\%$ died [7].

In contrast, Hsieh et al. found better performance with the age-adjusted SI and HaSI, both showing AUCs > 0.7 and statistically significant associations with ICU admission, intubation, and mortality (p < 0.001) [6].

Combining SI with other clinical parameters may improve its diagnostic utility. In this regard, Eldaboosy et al. proposed a combination of SI > 0.7 and $PaO_2/FiO_2 < 250$ (SI-PaFi), which demonstrated notable performance: an AUC of 0.89 for ICU admission and 0.90 for mortality, with sensitivity and specificity above 85%, outperforming APACHE II and CURB-65 scores [21]. This approach may represent a promising modification for evaluating patients with COVID-19 and warrants further investigation.

These mixed findings are reflected in the broader literatura. A recent systematic review and meta-analysis by Alsagaff et al. (n = 4,576) showed that an elevated SI was associated with increased mortality (OR: 7.52; 95% CI: 3.72–15.19; p < 0.00001), with an AUC of up to 0.84, sensitivity of 78%, and specificity of 76.8%. It was also associated with ICU admission (OR: 5.81; 95% CI: 1.18–28.58; p = 0.03). However, the main limitation of this meta-analysis was the high heterogeneity among included studies (>95%), which restricts the generalizability of its conclusions [8]. Five studies were conducted in Turkey, with others in Saudi Arabia, Taiwan, and the Netherlands; none were from Latin America or similar

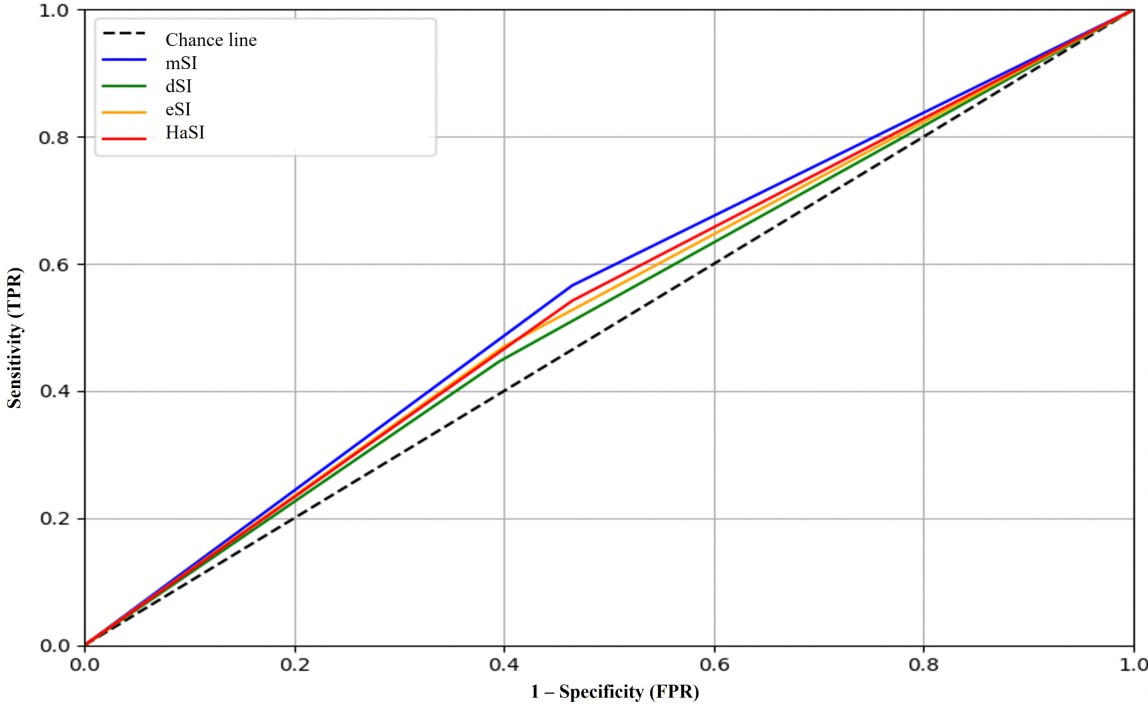

**Fig 2. ROC analysis for each shock index variant predicting ICU admission.** Predictive performance of mSI, dSI, eSI, and HaSI for ICU admission.

middle-income settings. The studies used varying SI cutoffs (0.6 to 1.0), different outcome definitions (in-hospital, 14-day, 30-day, and 90-day mortality), and one study defined high SI as >0.7 combined with hypoxemia (PaO2/FiO2 < 250) [21], creating a composite index rather than evaluating SI alone. Notably, Van Rensen et al., also included in that meta-analysis, showed results consistent with ours. The substantial heterogeneity and the inclusion of studies with widely varying methodologies suggest that the pooled estimates may not be uniformly applicable across all clinical contexts, particularly in settings with different patient characteristics and healthcare resources like ours.

Several factors specific to our population and setting may explain SI's limited discriminative capacity in our study. First, SI was originally designed for hemorrhagic and septic shock, where the heart rate-to-blood pressure ratio reflects intravascular volume depletion and compensatory cardiovascular responses. However, in moderate COVID-19 pneumonia, the primary pathophysiology is hypoxemic respiratory failure with initially preserved hemodynamic; patients may maintain relatively normal vital signs while developing progressive hypoxemia, pulmonary microthrombi, and inflammatory responses that do not necessarily manifest as classic shock physiology at emergency department presentation.

Second, our patients received standardized evidence-based COVID-19 treatment protocols including corticosteroids, anticoagulation, and early respiratory support. These measures may have altered the natural hemodynamic trajectory of the disease and diminished the predictive value of initial vital sign-based indices.

Third, as a tertiary care center in a middle-income country, ICU admission decisions may be influenced by resource constraints and bed availability, introducing selection pressures that differ from settings with greater critical care capacity. These contextual factors suggest that SI performance is highly setting-dependent, and our findings underscore the critical importance of prospective, context-specific validation before implementing any triage tool in clinical practice, particularly in Latin American healthcare systems and specific COVID-19 severity subgroups where data remain limited. Rather than contradicting the existing literature, our study contributes to understanding the external validity limitations of SI and

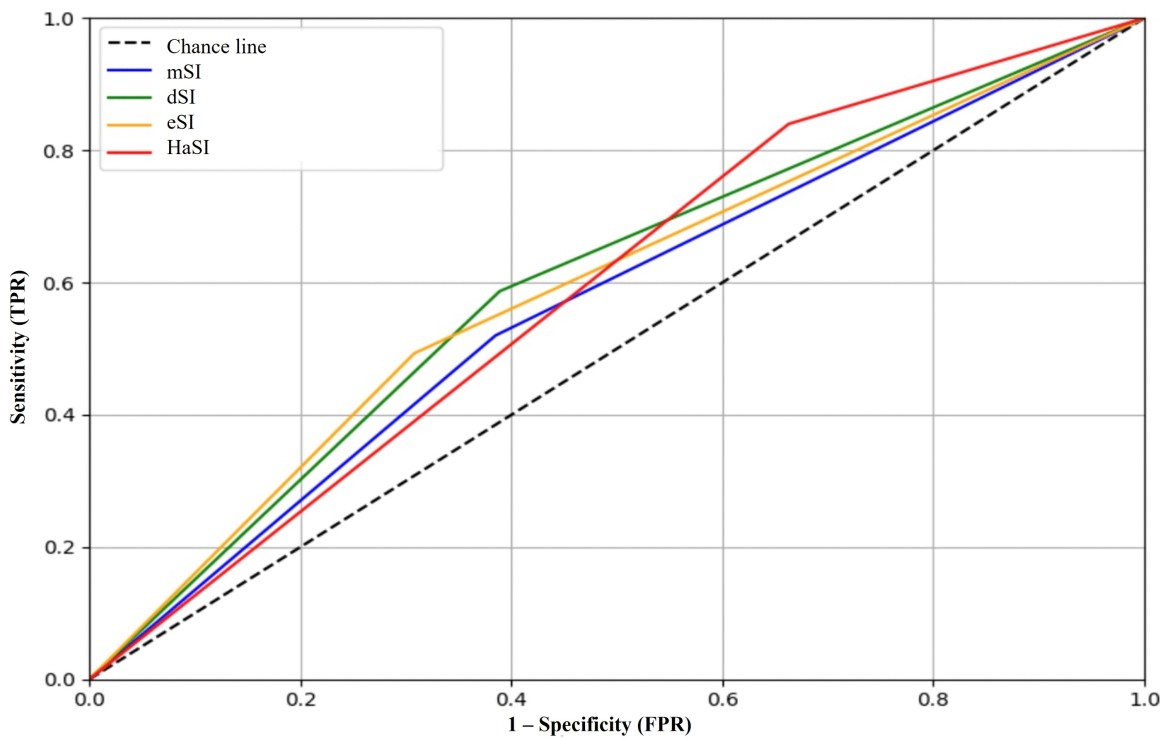

**Fig 3. ROC analysis each shock index variant predicting hospital length stay.** Predictive performance of mSI, dSI, eSI, and HaSI for hospital length of stay.

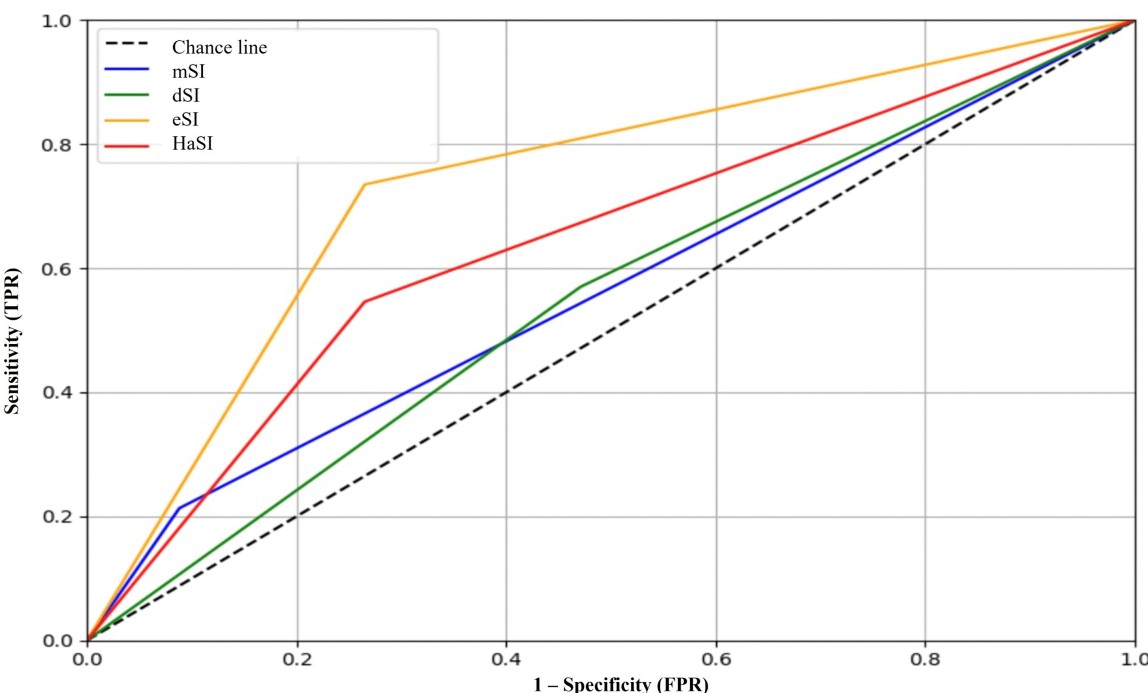

**Fig 4. ROC analysis each shock index variant predicting in-hospital mortality.** Predictive performance of mSI, dSI, eSI, and HaSI for in-hospital mortality.

highlights that pooled estimates from heterogeneous studies should be interpreted cautiously and validated locally before clinical implementation.

This study has several limitations that should be considered when interpreting its findings. First, it is a retrospective, single-center study, which limits the representativeness of our cohort and hinders the generalizability of the results to other institutions with different population characteristics, available resources, or clinical management practices. Furthermore, our final sample size was smaller than initially estimated, which may have reduced the statistical power to detect significant differences between shock index variants. This limited sample size may also have contributed to the modest AUCs observed and affects the precision of the diagnostic parameters calculated, particularly in subgroup analyses.

It is important to note that the data analyzed refer exclusively to the point of hospital admission. Most patients did not present with severe hypoxemia or require immediate support, which may explain the limited performance of the indices. Clinical instability may develop in later stages of the disease, suggesting the need for serial monitoring of the shock index and its variants during hospitalization, as well as the development of multivariable models that integrate the shock index with other clinical and laboratory markers to compare their performance against conventional prognostic tools.

## Conclusion

Early identification of patients at risk for complications in moderate COVID-19 is crucial for optimizing hospital resources, particularly in settings with high healthcare demand. Our findings demonstrate that the shock index and its variants have limited discriminative ability as standalone predictors for mortality, ICU admission, and hospital length of stay in patients with moderate COVID-19. While some evidence suggests potential benefits of combining the shock index with other clinical parameters, significant heterogeneity across studies limits the generalizability of these findings. Our results indicate that shock index variants should not be relied upon in isolation for risk stratification in moderate COVID-19. Future research should focus on developing and prospectively validating multivariable prediction models that incorporate multiple clinical, laboratory, and imaging biomarkers, with careful attention to local contexts and patient populations.

## Author contributions

**Conceptualization:** Annyi Tatiana Belalcazar, Valeria Monroy Lasso, José Darío Álvarez Herazo, Ana Clarete, Roger Figueroa-Paz, Duban Maya-Portillo, Julio Diez-Sepúlveda.

**Data curation:** Ana Clarete, Roger Figueroa-Paz, Julio Diez-Sepúlveda.

**Formal analysis:** Ana Clarete, Roger Figueroa-Paz.

**Investigation:** Annyi Tatiana Belalcazar, Valeria Monroy Lasso, José Darío Álvarez Herazo, Ana Clarete, Roger Figueroa-Paz, Duban Maya-Portillo, Julio Diez-Sepúlveda.

**Methodology:** Ana Clarete, Roger Figueroa-Paz.

**Project administration:** Ana Clarete, Julio Diez-Sepúlveda.

**Supervision:** Ana Clarete.

**Validation:** Ana Clarete, Duban Maya-Portillo, Julio Diez-Sepúlveda.

**Visualization:** Roger Figueroa-Paz.

**Writing – original draft:** Annyi Tatiana Belalcazar, Valeria Monroy Lasso, José Darío Álvarez Herazo, Ana Clarete, Roger Figueroa-Paz, Duban Maya-Portillo, Julio Diez-Sepúlveda.

**Writing – review & editing:** Annyi Tatiana Belalcazar, Valeria Monroy Lasso, José Darío Álvarez Herazo, Ana Clarete, Roger Figueroa-Paz, Duban Maya-Portillo, Julio Diez-Sepúlveda.

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
