## [Decision Letter · Decision Letter 0]

15 Oct 2025

Dear Dr. Clarete,

Thank you for submitting your manuscript to PLOS ONE. After careful consideration, we feel that it has merit but does not fully meet PLOS ONE’s publication criteria as it currently stands. Therefore, we invite you to submit a revised version of the manuscript that addresses the points raised during the review process.

We look forward to receiving your revised manuscript.

Kind regards,

Armaan Jamal

Guest Editor

PLOS ONE

Journal Requirements:

Reviewers' comments:

Reviewer's Responses to Questions

**Comments to the Author**

1. Is the manuscript technically sound, and do the data support the conclusions?

Reviewer #1: Partly

Reviewer #2: Yes

2. Has the statistical analysis been performed appropriately and rigorously?

Reviewer #1: Yes

Reviewer #2: Yes

3. Have the authors made all data underlying the findings in their manuscript fully available?

Reviewer #1: No

Reviewer #2: Yes

4. Is the manuscript presented in an intelligible fashion and written in standard English?

Reviewer #1: Yes

Reviewer #2: Yes

Reviewer #1: I would like to congratulate the author on the work presented. The article is concise and addresses an important problem in the clinical setting. The validation of an easy-to-use index can improve patient care and reduce undesirable outcomes.

However, the study has several flaws that undermine the quality of the presentation. I have several suggestions to improve the clarity of the study and address possible biases.

Introduction: The introduction is concise, direct, and well written; however, there is some information that is required and would improve the study's contextualization and help the reader understand the study's aim.

The authors, in lines 72 and 73, point out that various predictive models were developed to identify complications for covid, however, APACHE II and SOFA were not developed for covid assessment. They were used, and SOFA is one of the more reliable severity and mortality models for covid. Several other methods were developed, and some publications present a review of those scores and their reliability. I can suggest the use of Professor Appel work (Appel KS, Geisler R, Maier D, Miljukov O, Hopff SM, Vehreschild JJ. A systematic review of predictor composition, outcomes, risk of bias, and validation of COVID-19 prognostic scores. Clin Infect Dis [Internet]. 2024 Apr 10;78(4):889–99.)

In line 75, the authors state that few studies have examined the role of hemodynamic parameters and indicate the shock index as an example. The work previously indicated shows that blood pressure evaluation is present in 12,4% of the models. Heart rate is in Professor Marcolino’s work as well (10.1016/j.ijid.2021.07.049).

It is important to note that SI was not used in these studies, so I suggest that the author focus this paragraph on the lack of use of the shock index and its reliability and point out the importance of this index in clinical practice before covid and nowadays in the need for a quick assessment of covid patients.

In lines 85 through 87, the authors point out the study justification. However, new information is presented, like eSI and HaSI. I suggest that these modifications should be present and explained previously, and why they could outperform or underperform SI.

Methods: The methods must be more descriptive to improve reproducibility.

In line 102, the authors indicated that a confirmed diagnosis of moderate covid was one of the inclusion criteria. I suggest adding a reference in the text from the NIH criteria.

In the methods section, there is no description of the variables collected, no description of the outcomes measured, and how they were measured.

Items 7, 8, 9, 10, and 11 from STROBE are missing from the methods section. There is no way to understand how the authors measured the variables collected.

Results: The results are presented concisely and clearly. It lacks a fluxogram indicating patients’ inclusion and exclusion criteria, showing the path from population to sample. The ROC figures are in low resolution.

Discussion: The discussion needs to be improved; other articles can be added to discuss the results and the differences observed from the data presented to the literature.

Merge line 230 with the paragraph from line 228.

The paragraph from line 236 should cite the model that had high specificity, since it is a core topic.

The paragraphs from lines 241 through 256 show that the SI model has different performances in the literature in predicting mortality, severity, and other outcomes. The authors present that difference, but they did not discuss it. They present a meta-analysis that showed high AUROC, but do not discuss the studies in this meta-analysis, the differences between the studies, where they were developed, and why there is a discrepancy in the literature. Is the metanalisys pointing in the wrong direction, or is there any characteristic from the presented study that reduces the performance of SI as a predictive model?

This information is crucial to the reader to understand the importance of the study and why the literature differs from or points to other direction.

In the limitations, the author presents important information that should be in the methods section. This study lacks a better description of how it was performed to enhance its quality.

Conclusion: The authors still defend the use of SI regarding the results, and didn’t cite the meta-analysis to support this suggestion.

Reviewer #2: I have read with great interest the article “Rethinking emergency risk assessment: a single-center look at shock index and its variants in COVID-19.” The topic is highly relevant, as early identification of risk in patients with COVID-19 remains a clinical priority. However, I would like to make a few observations for consideration.

First, the study is described as analytical, yet no sample size estimation is reported, and the primary outcomes are not clearly defined in the Methods section. This information is essential for assessing both the internal validity and the generalizability of the results.

Second, although the concept of using the shock index and its variants is clinically interesting, the timing of measurement appears questionable. In the early phase of COVID-19, most patients are hemodynamically stable; the predominant problem is hypoxemia rather than shock. Consequently, at hospital admission, the shock index may not adequately capture the primary pathophysiological process leading to organ dysfunction and mortality in these patients. The authors should provide a clear rationale for evaluating this index at admission instead of at later stages of clinical deterioration.

Overall, this is a promising study that addresses an important question. However, clarification regarding sample size, outcome definition, and the timing of measurement would strengthen the conclusions and clinical applicability of the findings.

**Do you want your identity to be public for this peer review?** For information about this choice, including consent withdrawal, please see our Privacy Policy

Reviewer #1: **Yes:** Rafael Lima Rodrigues de Carvalho

Reviewer #2: **Yes:** David Rene Rodríguez Lima

---

## [Author Response · Author response to Decision Letter 1]

12 Dec 2025

Reviewer #1:

1. Introduction: The introduction is concise, direct, and well written; however, there is some information that is required and would improve the study's contextualization and help the reader understand the study's aim.

The authors, in lines 72 and 73, point out that various predictive models were developed to identify complications for covid, however, APACHE II and SOFA were not developed for covid assessment. They were used, and SOFA is one of the more reliable severity and mortality models for covid. Several other methods were developed, and some publications present a review of those scores and their reliability. I can suggest the use of Professor Appel work (Appel KS, Geisler R, Maier D, Miljukov O, Hopff SM, Vehreschild JJ. A systematic review of predictor composition, outcomes, risk of bias, and validation of COVID-19 prognostic scores. Clin Infect Dis [Internet]. 2024 Apr 10;78(4):889–99.)

R/ We have made the suggested correction and clarified that although APACHE II and SOFA were not specifically developed for COVID-19, they have been widely used, and SOFA remains one of the more reliable models for assessing severity and mortality in COVID-19 patients. Additionally, we have included the reference recommended by the reviewer (Appel KS et al., 2024) to support this point.

2. In line 75, the authors state that few studies have examined the role of hemodynamic parameters and indicate the shock index as an example. The work previously indicated shows that blood pressure evaluation is present in 12,4% of the models. Heart rate is in Professor Marcolino’s work as well (10.1016/j.ijid.2021.07.049).

It is important to note that SI was not used in these studies, so I suggest that the author focus this paragraph on the lack of use of the shock index and its reliability and point out the importance of this index in clinical practice before covid and nowadays in the need for a quick assessment of covid patients.

R/ We have revised the paragraph to indicate that some predictive models do include hemodynamic parameters, citing the work of Professor Marcolino and the systematic review by Professor Appel as examples. At the same time, we emphasized that the shock index (SI) specifically has not been used in these studies. We also highlighted the clinical importance of SI, both prior to COVID-19 and currently, as a rapid and reliable tool for early patient assessment.

3. In lines 85 through 87, the authors point out the study justification. However, new information is presented, like eSI and HaSI. I suggest that these modifications should be present and explained previously, and why they could outperform or underperform SI.

R/ We have updated the manuscript so that the variables of the shock index (SI), including its modifications such as eSI and HaSI, are now presented and explained prior to the study justification. This allows readers to understand these modifications and their potential to outperform or underperform the traditional SI before introducing the rationale for the study.

4. Methods: The methods must be more descriptive to improve reproducibility.

In line 102, the authors indicated that a confirmed diagnosis of moderate covid was one of the inclusion criteria. I suggest adding a reference in the text from the NIH criteria.

R/ A clearer description of the NIH criteria for moderate COVID-19 has been provided in the Methods section, and the corresponding reference has been added to improve reproducibility.

5. In the methods section, there is no description of the variables collected, no description of the outcomes measured, and how they were measured.

Items 7, 8, 9, 10, and 11 from STROBE are missing from the methods section. There is no way to understand how the authors measured the variables collected.

R/ Thank you for your suggestion. We have tried to supplement the methods section with the information you requested and that requested in the STROBE guide. With regard to item 11, please refer to the statistical analysis section for information on the handling of quantitative variables.

6. Results: The results are presented concisely and clearly. It lacks a fluxogram indicating patients’ inclusion and exclusion criteria, showing the path from population to sample. The ROC figures are in low resolution.

R/ We have created the patient selection flowchart (Fig 1) and revised the patient selection section as requested, and we have improved the resolution of the ROC figures. We have also clarified the exclusion criteria: while our original protocol included patients managed exclusively via telemedicine as an exclusion criterion, a review at the time of filtering our registry during the study period showed that no patients were actually attended entirely through telemedicine (n = 0). Therefore, this criterion has been removed from the Methods section to reflect only the exclusions that were actually applied.

7. Discussion: The discussion needs to be improved; other articles can be added to discuss the results and the differences observed from the data presented to the literature.

Merge line 230 with the paragraph from line 228.

R/ According to your suggestion, we have merged line 230 with the paragraph starting at line 228.

8. The paragraph from line 236 should cite the model that had high specificity, since it is a core topic.

R/ We have made the modification as suggested; the model with high specificity is now mentioned and properly referenced. This information has been incorporated in the revised manuscript.

9. The paragraphs from lines 241 through 256 show that the SI model has different performances in the literature in predicting mortality, severity, and other outcomes. The authors present that difference, but they did not discuss it. They present a meta-analysis that showed high AUROC, but do not discuss the studies in this meta-analysis, the differences between the studies, where they were developed, and why there is a discrepancy in the literature. Is the metanalisys pointing in the wrong direction, or is there any characteristic from the presented study that reduces the performance of SI as a predictive model?

This information is crucial to the reader to understand the importance of the study and why the literature differs from or points to other direction.

R/ We have included a discussion of the studies cited, highlighting differences in study populations, settings, and methodologies, which may contribute to the variability in SI performance reported in the literature.

Concerning the apparent discrepancy between our findings and those reported in the meta-analysis by Alsagaff et al., we have revised the relevant paragraphs to facilitate a more thorough discussion. Rather than suggesting that the meta-analysis is "pointing in the wrong direction," our study emphasizes that SI performance is highly context-dependent and that pooled estimates from heterogeneous studies should be interpreted cautiously. Our work highlights the importance of conducting specific validation studies before adopting SI or any triage tool in clinical practice, particularly in middle-income countries and specific subgroups of severe cases of COVD-19 where data are limited.

10. In the limitations, the author presents important information that should be in the methods section. This study lacks a better description of how it was performed to enhance its quality.

R/ A more detailed description of the methods section has been created. This includes migrating information from the limitations section to the methods section.

11. Conclusion: The authors still defend the use of SI regarding the results, and didn’t cite the meta-analysis to support this suggestion.

R/ We have revised the conclusion, taking into account the studies included in the meta-analysis. Our conclusion emphasizes the limited discriminative ability of the shock index and its variants in moderate COVID-19 illness, while acknowledging the heterogeneity of results reported in the literature. This ensures that the conclusion reflects the evidence available and provides a cautious interpretation of SI performance, without suggesting it should be used in isolation for risk stratification.

Reviewer #2:

1. First, the study is described as analytical, yet no sample size estimation is reported, and the primary outcomes are not clearly defined in the Methods section. This information is essential for assessing both the internal validity and the generalizability of the results.

R/ We agree that clarity regarding the analytical nature of the study is important. The study is described as analytical because it goes beyond simple descriptive reporting; it evaluates associations and predictive performance of the shock index (SI) and its variants for clinically relevant outcomes (ICU admission, hospital stay, and in-hospital mortality). Specifically, we applied inferential and diagnostic accuracy analyses, including ROC curves, AUC estimation, and calculation of predictive metrics (PPV, NPV, likelihood ratios), which are hallmarks of analytical observational studies. Regarding your other points:

• Sample size estimation: As stated in the Methods section, no formal sample size calculation was performed because the study included all eligible patients from the RECOVID registry during the study period, ensuring comprehensive coverage of the target population.

• The primary outcomes have been clearly defined in the updated Methods section. These outcomes include ICU admission, hospital stay and in-hospital mortality.

These aspects have been clarified in the manuscript to enhance internal validity and transparency.

2. Second, although the concept of using the shock index and its variants is clinically interesting, the timing of measurement appears questionable. In the early phase of COVID-19, most patients are hemodynamically stable; the predominant problem is hypoxemia rather than shock. Consequently, at hospital admission, the shock index may not adequately capture the primary pathophysiological process leading to organ dysfunction and mortality in these patients. The authors should provide a clear rationale for evaluating this index at admission instead of at later stages of clinical deterioration.

R/ We thank the reviewer for this important observation. The use of the SI at hospital admission is justified by its simplicity, rapid calculation using standard vital signs, and applicability in emergency settings, particularly in resource-limited contexts. In our study, the data analyzed refer exclusively to patients with moderate COVID-19 illness at the point of hospital admission. Most patients did not present with severe hypoxemia or require immediate support, which may explain the limited performance of the indices in this population.

The SI allows for a quick, non-invasive assessment of hemodynamic status, which is crucial in high-demand emergency settings where timely decisions are needed. Measuring the SI at admission enables early identification of patients with subtle hemodynamic instability who may be at risk of deterioration or require ICU referral, before severe complications develop. While the early phase of COVID-19 is often dominated by hypoxemia, SI can detect early cardiovascular instability that may precede or coexist with hypoxemia.

We acknowledge that clinical instability may develop later, which supports the potential value of serial monitoring and the integration of SI with other clinical and laboratory parameters to enhance predictive performance against conventional prognostic tools, as reflected in our conclusion.

---

## [Editor Report · Decision Letter 1]

11 Jan 2026

Rethinking emergency risk assessment: a single-center look at shock index and its variants in COVID-19

PONE-D-25-37663R1

Dear Dr. Clarete,

We’re pleased to inform you that your manuscript has been judged scientifically suitable for publication and will be formally accepted for publication once it meets all outstanding technical requirements.

Kind regards,

Armaan Jamal

Guest Editor

PLOS One

---

## [Editor Report · Acceptance letter]

PONE-D-25-37663R1

PLOS One

Dear Dr. Clarete,

I'm pleased to inform you that your manuscript has been deemed suitable for publication in PLOS One. Congratulations! Your manuscript is now being handed over to our production team.

Kind regards,

on behalf of

Mr. Armaan Jamal

Guest Editor

PLOS One